

# Landsifier v1.0: a Python library to estimate likely triggers of mapped landslides

Kamal Rana[1,2,3], Nishant Malik[4], and Ugur Ozturk[1,3]

[1]Helmholtz-Centre Potsdam–GFZ German Research Centre for Geosciences, Potsdam, Germany.
[2]Chester F. Carlson Center for Imaging Science, Rochester Institute of Technology, Rochester, NY, USA.
[3]Institute of Environmental Science and Geography, University of Potsdam, Potsdam, Germany.
[4]School of Mathematical Sciences, Rochester Institute of Technology, Rochester, NY, USA.

**Correspondence:** Kamal Rana (kr7843@rit.edu)

**Abstract.**

Landslide hazard models aim at mitigating landslide impact by providing probabilistic forecasting, and the accuracy of these models hinges on landslide databases for model training and testing. Landslide databases at times lack information on the underlying triggering mechanism, making these inventories almost unusable in hazard models. We developed a Python-based library, `landsifier`, that contains three different Machine-Learning frameworks for assessing the likely triggering mechanisms of individual landslides or entire inventories based on landslide geometry. Two of these methods only use the 2D landslide planforms, and the third utilizes the 3D shape of landslides relying on an underlying Digital Elevation Model (DEM). The base method extracts geometric properties of landslide polygons as a feature space for the shallow learner—Random Forest (RF). An alternative method extracts topological properties of 3D landslides through Topological Data Analysis (TDA) and then feeds these properties as a feature space to the Random Forest classifier. The last framework relies on landslide-planform images as an input for the deep learning algorithm—Convolutional Neural Network (CNN). We tested all three interchangeable methods on several inventories with known triggers spread over the Japanese archipelago. To demonstrate the effectiveness of developed methods, we used two testing configurations. The first configuration merges all the available data for the k-fold cross-validation, whereas the second configuration excludes one inventory during the training phase to use as the sole testing inventory. Classification accuracies for different testing schemes vary between 70% and 95%. Finally, we implemented the three methods on an inventory without any triggering information to showcase a real-world application.

## 1 Introduction

Landslides are gravitational movements of rock and debris that pose a severe threat to the human environment. Hazard models are developed to forecast landslides or to aid in understanding landslide processes to mitigate their undesired consequences (Lombardo et al., 2020). These models commonly rely on mapped landslides to assess the relevant landslide causes in combination with landslide triggers, i.e., earthquake and rainfall (Lombardo and Tanyas, 2021; Ozturk et al., 2021; Marin et al., 2020). However, many historical landslide inventories lack information about the triggering mechanism decreasing their potential utility in models (Bíl et al., 2021; Martha et al., 2021). More recent semi-automated satellite-based landslide mappers





also often disregard the triggering information (Behling et al., 2014, 2016; Ghorbanzadeh et al., 2019), except the event-based
inventories—landslide mapping campaigns following a precursory triggering event such as a strong earthquake (Stumpf and
Kerle, 2011; Gorum et al., 2014). Using landslide inventories with missing triggers could introduce biases as it is possible to
accidentally use an earthquake-triggered inventory to assess rainfall-induced landslide hazards and vice-versa. Hence, classi-
fying the trigger of entire landslide inventories or mapped individual landslides would enhance the usability of newly acquired
and historical inventories in landslide models.

Landslides with the same trigger morphologically cluster, for example, covering narrowly the available statistical variability
of hillslope angles in a study region (e.g., Jones et al., 2021) and, thus, could have characteristic shapes reflecting their triggering
mechanism, for instance, by having similar area and perimeter ratio, or size (Taylor et al., 2018; Samia et al., 2017). We
developed a binary classifier that groups landslides either as earthquake-triggered or rainfall-induced based on this hypothesis
(Rana et al., 2021). This initial model demonstrated that the landslides with an identical trigger indeed exhibit similar geometric
properties. Thus, finding the trigger of landslides is a classification problem, and one can employ machine learning tools to carry
out automated classification of landslide triggers. In each classification problem, the principal idea is to construct a classifier
based on training samples and evaluate its performance on testing samples. The classifier predicts the class $y$ corresponding
to the input sample $x$. These input samples $x$ can be one-dimensional vectors or images; for instance, in a soil classification
problem (e.g., Bhattacharya and Solomatine, 2006), $x$ is a one-dimensional vector, and in any image classification problem, $x$
is an image (2D or multi-dimensional matrix) (Domingos, 2012).

Our preliminary model (Rana et al., 2021) can classify landslide triggers by only using the geometric properties of landslide
polygons. Here, we introduce two additional methods for landslide trigger classification. In one new method, we treated land-
slide polygons as images, and these images are fed as the sole predictor to a deep learner—Convolutional Neural Networks
(CNN). Treating landslide polygons as images eases the workflow as an image already resembles some of the geometric fea-
tures of the first method. Both these methods rely on two-dimensional (2D) landslide planforms, ignoring the three-dimensional
(3D) shapes of real-world landslides. In another approach, we included the 3D shapes of landslides by incorporating the el-
evation of landslides via a Digital Elevation Model (DEM). In this approach, we extracted the topological features of these
3D shapes using a recently developed technique known as Topological Data Analysis (TDA). These topology-based features
are input to the decision tree-based shallow learner as in the first method. We included the TDA-based model considering its
potential to handle other relevant classification problems in future versions of our tool, e.g., classifying landslide types (Cruden
and Varnes, 1996; Varnes, 1978). Above listed methods could be used independently following similar script streams.

This study also introduces a new Python library, landsifier, that classifies the trigger of landslides, individually or
as a whole, in an inventory, where the landslide source mechanism is undocumented. The library consists of three different
machine learning-based methods mentioned above; we elaborate on these methods in section 3. Various functionalities of
the library are described in Appendix B; where we also list several supporting functions to calculate landslide polygons'
geometric properties, convert landslide polygons' shape to a binary-scale image, download a Digital Elevation Model (DEM)
corresponding to inventory location, and evaluate the diagnostic performance of the final classification. To demonstrate the
efficacy of the developed methods, we apply each to six landslide inventories with known triggers spread over the Japanese



archipelago and document our findings in section 4. In section 6, we further highlight the weaknesses of each method to ease

choosing the suitable classifier for the various applications.

## 2   Data

In this work, we used seven landslide inventories spread over the Japanese archipelago (Figure 1). The trigger mechanism
of six out of seven landslide inventories are known (Figure 1a–f), whereas the last inventory has no documented triggering
information (Figure 1g). We use the last inventory to demonstrate the practical deployment of the final model as this case

represents the model's real-world usage. Out of six landslide inventories, three inventories are earthquake-triggered (Figure 1d–
f) that are associated with the 2018 $M_W 6.6$ Hokkaido Eastern Iburi (3256 landslides); the 2008 $M_W 6.9$ Iwate–Miyagi Nairiku
(4160 landslides), and the 2004 $M_W 6.6$ Niigata (8780 landslides). The remaining three are rainfall-induced (Figure 1a–c), and
these are associated with the 2017 Fukuoka-northern Kyushu torrential rainfall disaster (1924 landslides), the 2018 Saka-Japan
floods (2817 landslides), and Kumamoto inventory (5564 landslides) that is collected over 1992–2012—not associated with

any particular event.

The Geospatial Information Authority of Japan (GSI) is the source of the Hokkaido Eastern Iburi earthquake (September 2018), Fukuoka rainfall (July 2017), and Saka rainfall (July 2018) inventories. The source of the other two coseismic
inventories—Iwata and Niigata—is the global repository created by Schmitt et al. (2018). The remaining two inventories from
the Kumamoto region are provided by Japan's National Research Institute for Earth Science and Disaster Resilience (NIED).

The first inventory from Kumamoto is associated with rainfall (Figure 1b), whereas the second inventory is without any trig-
gering information (Figure 1g). From hereafter, we refer to this second inventory as "Kumamoto unspecified" (it consists of
612 landslides with unknown triggers).

The TDA-based method uses elevation data to obtain the 3D shapes of landslides from their 2D planforms. We use the
*Shuttle Radar Topography Mission* (SRTM) Digital Elevation Model (DEM) data that comes with a spatial resolution of

approximately 30 meters. The SRTM data is freely available from `https://www2.jpl.nasa.gov/srtm/` by manually
selecting the tiles which correspond to topographic quadrangles. Each tile covers 1 degree of both latitude and longitude
region. The `landsifier` library automatically downloads the corresponding tile(s) covering the region of the used landslide
inventory (explained further in Appendix B).

## 3   Methods

In our preliminary study (Rana et al., 2021), we introduced a method that can classify landslide triggers by only using geometric
features of landslide planforms. This initial model constitutes the first method in `landsifier` library, and for continuity, we
briefly describe it in section 3.1. In this paper, we further diversify our initial model and introduce two new methods, one based
on the topological features of 3D shapes of landslides computed using TDA; described in section 3.2. The other new method
**Figure 1.** The seven landslide inventories used in this work are spread over Japan, and their geographical locations are shown on the Country's map at the center of the figure. (a)–(g) Shows the subset of landslide polygons highlighted by red color on local hillshades. (a)–(c) Rainfall-induced inventories; (d)–(f) coseismic inventories; (g) undocumented "Kumamoto unspecified" inventory.




uses CNN to carry out an image-based classification of landslide triggers; see section 3.3. We anticipate that the variety of
methods and corresponding Python library presented here would allow researchers to perform this analysis seamlessly.

## 3.1   First method: geometric features based classification

In the first method, we used the geometric properties of 2D landslide polygons for the classification. We explored several
geometric properties of landslide polygons (e.g., Figure 2). Using a combination of feature selection methods and feature
importance analysis, for instance, removing highly correlated features, we choose the seven geometric properties of polygons
that lead to optimum results. These geometric features are area $A$, perimeter $P$, convex hull based measure $C_h = \dfrac{A}{A_c}$, where,
$A_c$ is the area of the convex hull fitted to the polygon (hereafter, we will refer $C_h$ as convex hull measure), the ratio of area to

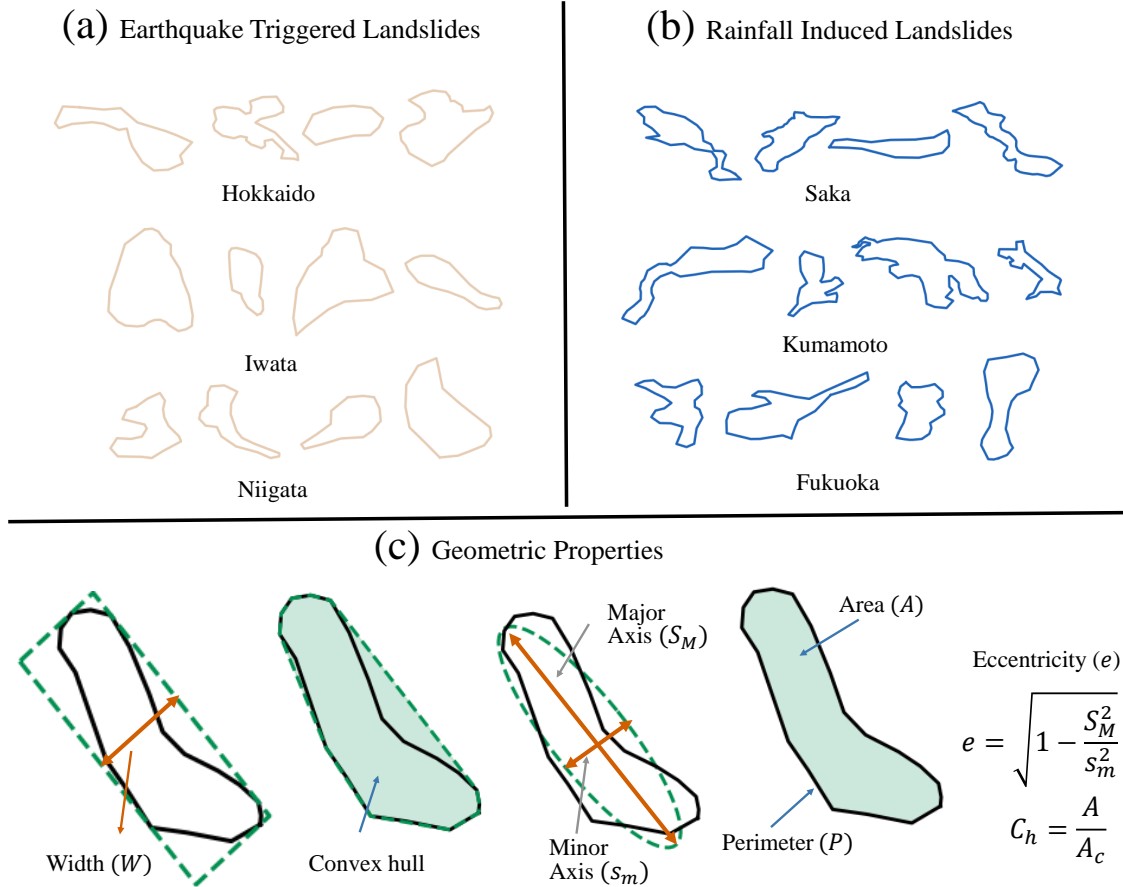

**Figure 2.** Sample landslide planforms from all six known triggered inventories (a) Earthquake triggered inventories, (b) Rainfall induced
inventories. (c) geometric properties of landslide polygon (from left to right): width ($W$) of the minimum area bounding box fitted to polygon,
convex hull based measure ($C_h$), minor($s_m$), and major axis($S_M$) of an ellipse fitted to polygon having area A and perimeter P, area ($A$) and
perimeter ($P$) of the polygon.




perimeter $\frac{A}{P}$, the width of the minimum area bounding box $W$, minor axis $s_m$, and eccentricity of the fitted ellipse $e$ having area $A$ and perimeter $P$. All these seven geometric features are calculated using the Python library, `shapely` (Gillies, 2013). The feature vector ($[A, P, C_h, W, s_m, \frac{A}{P}, e]$) is input variable to machine learning algorithm—random forest (described in the

Appendix A). Further details of the method can be found in (Rana et al., 2021).

### 3.2   Second method: topological features based classification

In the second method, we used the 3D shapes of landslides by incorporating the elevation data of the landslide regions. We extracted geometrical and topological properties of a landslides' 3D shapes using Topological Data Analysis (TDA) and then used these properties as a feature space for the machine learning algorithm—random forest (described in the Appendix A). We

converted the 2D landslide polygons to 3D landslide polygons using interpolation of 30 meters' elevation data (DEM) around the bounding box of landslides. We took only the elevation data within the landslide polygons to preserve the geometric shape of the landslides (Figure 3). We explored various TDA features to quantify the 3D shapes of landslides using the Python library, `giotto-tda` (Tauzin et al., 2021). Using random forest feature importance analysis, we selected the top ten most relevant features, as irrelevant features increase the complexity of the model and are ineffective in improving the classification results.

These selected relevant features constitute the input variables for the random forest classifier.

Topological Data Analysis (TDA) provides a gamut of metrics to quantify the multidimensional shape of data by applying techniques of algebraic topology (Carlsson, 2009). These metrics could also serve as a feature space for machine learning algorithms to solve classification problems, e.g., the classification of manifolds or complex geometric shapes. The central idea of TDA is persistent homology that identifies persistent geometric features in the data; it uses simplicial complexes to

extract topological features from the point cloud data. A simplicial complex is a collection of simplexes and building blocks of higher dimensional counterparts of a graph. For example, a point is a 0-dimensional simplex, an edge which is a connection between two points is 1-dimensional simplex, a filled triangle formed by connecting three non-linear points is a 2-dimensional simplex. In general, an n-dimensional simplex is formed by connecting n+1 affinely independent points (Munch, 2017; Garin and Tauzin, 2019).

Generally, in TDA, one constructs a simplicial complex by the Vietoris-Rips complex method, where one chooses a parameter $\epsilon > 0$ to find the structure present in the data. For each pair of points $(x, y)$ in the point cloud data, add an edge between $x$ and $y$ if euclidean distance ($d$) between $x$ and $y$ is less than $\epsilon$. For a $n$-dimensional simplex, distance between each pair of $n + 1$ affinely independent points should be less than $\epsilon$ ($d(x, y) < \epsilon$). Each value of $\epsilon$ provides a set of simplexes representing a data structure. Different values of $\epsilon$ could lead to a different structure in data. To get the complete information about the

structures present in the data, all the possible values of $\epsilon$ are used, creating a sequence of simplicial complex (this process is called filtration, Figure 4a-g).

Homology measures particular structures present in the data providing valuable information about the geometrical and topological properties of the data. For example 0-dimensional homology captures connected components or clusters, 1-dimensional homology measures loops, 2-dimensional homology measures voids (Munch, 2017; Hensel et al., 2021). Structures like con-

nected components, holes, and voids originate (birth) and disappear (death) with a change in the value of $\epsilon$. A persistence





diagram, shown in Figure 4(h); documents the birth and death information of these structures. Using the birth and death information of clusters, holes, and voids present in the persistence diagram, we can calculate several topological features of the data. We used various topological features to quantify the shape of data such as persistence entropy, average lifetime, number of points, betti curve-based measure, persistence landscape curve-based measure, Wasserstein amplitude, Bottleneck amplitude, Heat kernel-based measure, and landscape image-based measure. Each topological metric considers different homology dimensions separately.

**Figure 3.** Sample $3D$ landslides from six known triggered inventories, (a) flow chart of conversion of $2D$ landslide planforms to $3D$ landslide shape. (b) Earthquake triggered $3D$ landslide samples, (c) rainfall induced landslide $3D$ samples. The $2D$ landslide planforms converted to $3D$ landslide shapes by using the elevation of landslides through a Digital Elevation Model (DEM).

The above mentioned topological features can be explained using two objects, one the set of $\{(b_i, d_i)\}_{i=1}^{i=N}$ birth-death pair in the persistence diagram; where $i$ and $N$ are the birth-death pair index and the total number of birth-death pairs respectively, and two the elements of lifetime vector $[l_i]_{i=1}^{i=N}$, calculated as difference between death and life of $(b_i, d_i)$ pair ($l_i = d_i - b_i$). Then the number of points is the length of the lifetime vector, whereas Wasserstein and Bottleneck amplitudes are $p$-norm and $\infty$-norm of lifetime vector, respectively. Average lifetime and persistence entropy are average and Shannon-entropy of lifetime vector.

Betti and persistence landscape curves based features are calculated from $p$-norm of discretized betti and persistence landscape curves. Betti curve is a function $B(\epsilon)$ that maps persistence diagram to an integer-valued curve, $B(\epsilon) : \mathbb{R} \to \mathbb{Z}$, it counts the number of (birth, death) pairs at $\epsilon$ that satisfy the condition $b_i < \epsilon < d_i$ (Garin and Tauzin, 2019). Whereas, persistence land-

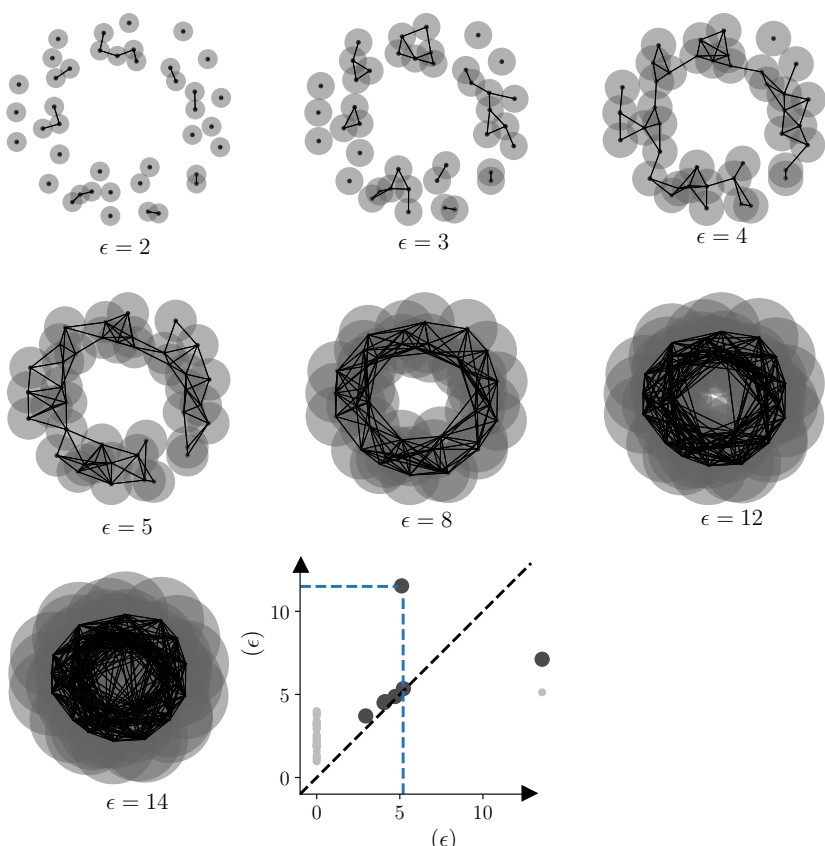

**Figure 4.** An example of using persistence homology: the data points are sampled from a noisy circle. (a)-(g) As the disk's radius increases ($\frac{\epsilon}{2}$), persistence homology captures various structures in the data. (h) The origin (birth) and disappearance (death) of loops and connected components is shown in the persistence diagram. The biggest loop in the noisy circle data is captured by the data points shown with the blue dotted line in (h).





scape curve is a function $\lambda(k, \epsilon) : \mathbb{R} \to \mathbb{R}_+$, where $\lambda(k, \epsilon) = k_{max}\{f_{b_i, d_i}(\epsilon)\}_{i=1}^{i=n}$, $k_{max}$ is $k$-th largest value of set of functions defined by $f_{b_i, d_i}(\epsilon) = max\{0, min(\epsilon - b_i, d_i - \epsilon)\}$ for each $(b_i, d_i)$ pair (Bubenik and Dłotko, 2017).

The heat kernel-based feature is calculated using the p-norm of the 2D function discretization obtained using the heat kernel on the persistence diagram. Heat kernel transforms the persistence diagram to a function on $\mathbb{R}^2$ obtained by placing a Gaussian
kernel with standard deviation $\sigma$ to each (birth, death) pair and negative of Gaussian kernel with same standard deviation in the mirror image of (birth, death) pairs across the diagonal (Reininghaus et al., 2015). Whereas persistence image-based measure is calculated using the p-norm of 2D function discretization obtained using the weighted Gaussian kernel on the birth-persistence diagram. Weighted Gaussian kernel transforms birth-persistence diagram to a function on $\mathbb{R}^2$ obtained by placing a weighted Gaussian kernel with standard deviation $\sigma$ to each (birth, death - birth) pair in birth-persistence diagram (Adams et al., 2017).
In the birth-persistence diagram, the y-axis represents the lifetime (death-birth) information of each (birth, death) pair.

### 3.3   Third method: image based classification

In the third method, we used landslide planform images as input to Convolutional Neural Networks (CNN) for the classification. We converted landslide polygons into binary images in a way that preserves the relative shape and structure of the polygons (Figure 5). Then using CNN for landslide triggers classification is straightforward via a simple CNN architecture with 3
convolutional layers and 2 fully connected layers. The input to CNN is a $64 \times 64$ binary pixel image, and the output is the probability of the input image belonging to one of the landslide trigger classes.

Convolutional Neural Networks (CNNs) are a class of artificial neural networks that are effective for various applications, such as image classification and object detection (Li et al., 2014; Guo et al., 2017; Albawi et al., 2017). The CNN architecture for classification problems consists of the input, hidden, and output layers (as shown in Figure 6). The input layer consists of
the input data to CNN, an image of a landslide polygon in our application. The hidden layer primarily contains convolutional layers, max-pooling, and fully connected layers. Finally, the output layer provides the probability of input data belonging to an output class—rainfall-induced or coseismic.

Convolutional layers are the fundamental component of CNN that uses kernels (matrix of learnable parameters) to perform convolutions operations on the input. The resulting output of the convolution operation is called a feature map that learns the
feature representation of the input data (Yamashita et al., 2018). Each neuron in a feature map captures the antecedent layer's local characteristics by convolution of kernels with the previous layer's feature maps (Guo et al., 2017). However, increasing convolutional layers could lead to over-parametrization and increase model complexity and, thus, over-fitting. One of the ways to avoid the issue is to use pooling layers that reduce feature maps dimension and the number of neurons in the output layer of CNN's (Yamashita et al., 2018; Guo et al., 2017). We used max-pooling layers of $n \times n$ ($n = 2$) size that takes a patch of
size $n \times n$ from a feature map and produces one-value corresponding to that patch, and the pooling layer itself is free from parameters (Li et al., 2014).

Activation functions in CNN's capture the non-linear relationship between the input data and its output class. We used ReLu for the hidden layer neurons activation functions as past studies have proved that ReLu improves classification results and learning speed (Li et al., 2014; Krizhevsky et al., 2012). The output of ReLu activation function is $f(x) = max(0, x)$, here




$x$ means the output of a neuron (Li et al., 2014). For the output layer, we used the softmax activation function. The softmax activation function calculates the output probabilities of the input sample belonging to each class in the last layer of CNN. The class probabilities are calculated as

$$P_i = \frac{\exp z_i}{\sum_{j=1}^{j=m} \exp z_j},\tag{1}$$

where $z_i$ is the output from last layer of CNN corresponding to $i$ class and $m$ is the number of classes (in our case, $m = 2$).

Fully connected layers (FCC) work as a classification layer for CNNs and comes after the convolutional layers. All layers in FCC are fully connected which means each neuron in a layer is connected to every neuron in the next layer of FCC (Albawi et al., 2017; Guo et al., 2017). In classification problems, the last layer of the FCC layer gives the probabilities of the input image to belong to one of the output classes with the help of the softmax activation function (Eq. 1). The output predicted probabilities of the input sample are used in a loss function that evaluates how well the model works for classifying the class

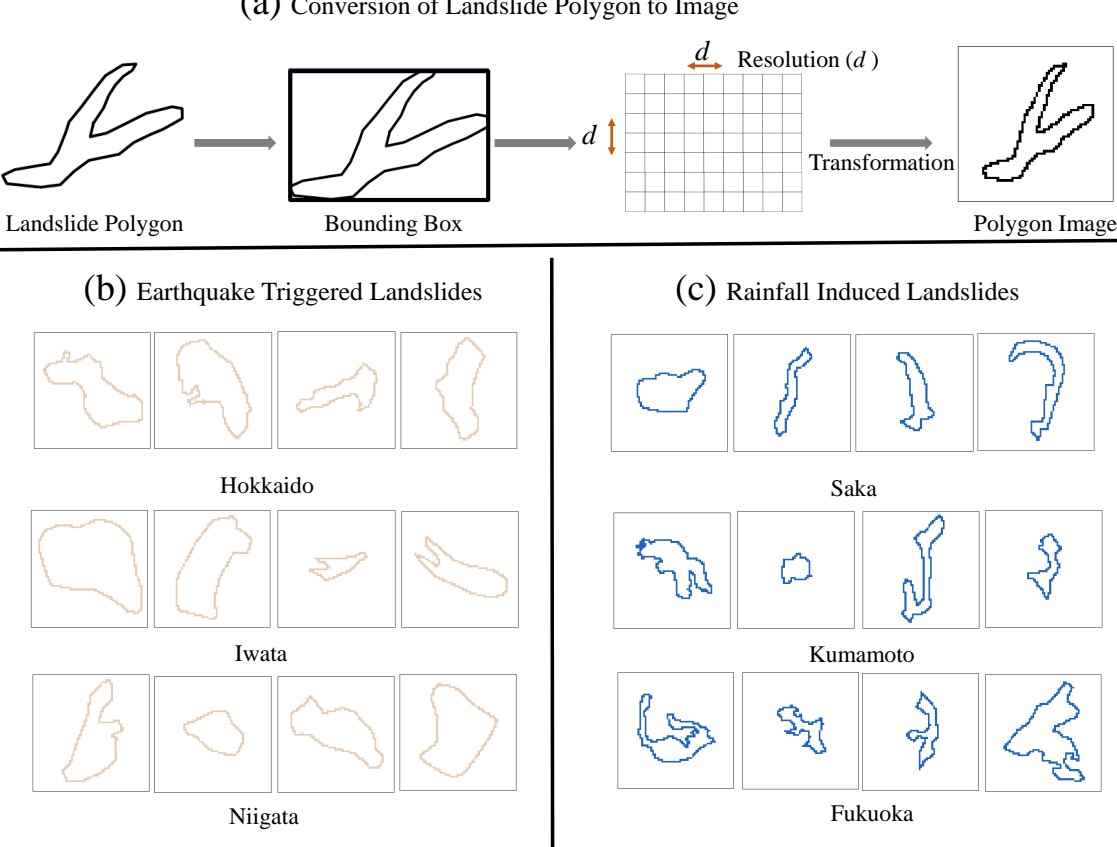

**Figure 5.** Sample input images for the image-based classification. (a) Flow chart of converting landslide planforms to a landslide polygon image. (b) Earthquake-triggered landslide image samples. (c) Rainfall-induced landslide image samples.





of the input image dataset. We used the cross-entropy loss function that measures the difference between actual and predicted probability distribution. The cross-entropy loss function for a sample is defined as: $-\sum_{i=1}^{i=m} y_i log(\hat{y_i})$, where $m$ is the total number of classes, $y_i$ $(\hat{y_i})$ is actual (predicted) probability corresponding to class $i$. If $i$ is actual class of the input sample then $y_i = 1$, otherwise $y_i = 0$. In the case of binary classification $m = 2$. The sample's output probabilities are a function of parameters used in convolution kernels and FCC layers to connect neurons in one layer to the next layer. These parameters are

altered iteratively using the back-propagation algorithm and stochastic gradient method to increase the probability of samples belonging to the actual class and thus, minimize the loss (Aurisano et al., 2016).

## 4   Landsifier model evaluation

We used two different test and training set split configurations to evaluate the efficacy of our methods. In the first configuration, we combined all the inventories with known triggers in the data and then split the data set into various training and testing sets.

In the second split configuration, we use all the possible combinations to train the algorithm on five inventories and test it on the sixth inventory. Note that there are seven inventories in the data set analyzed here, and six of these have known triggers. The analysis of this seventh inventory (Kumamoto unspecified) with unknown triggers is presented in the section 6.

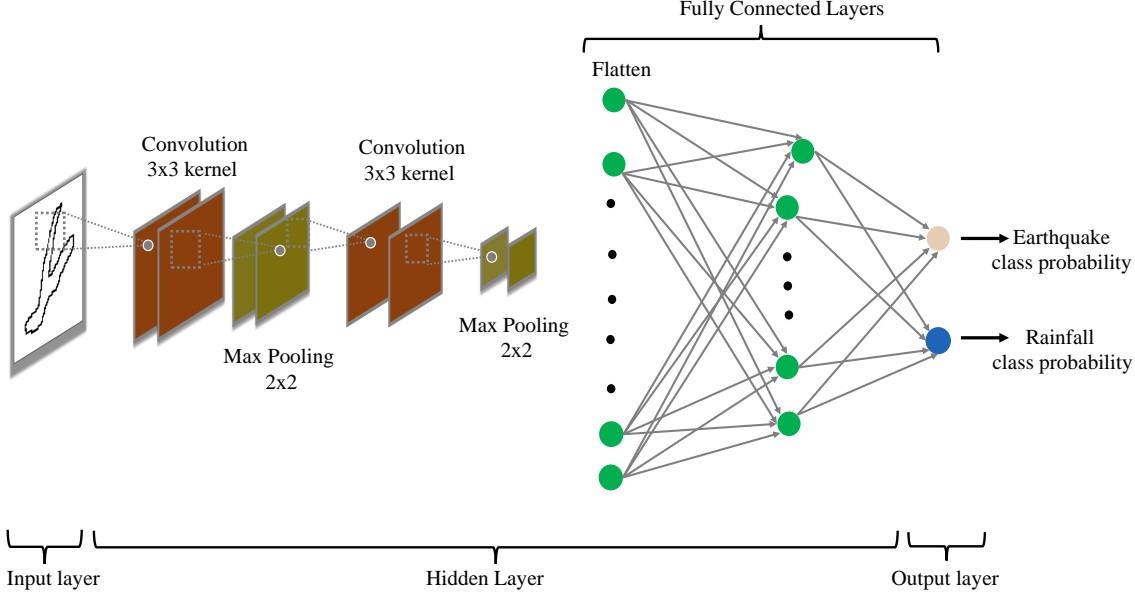

**Figure 6.** The figure shows the Convolutions Neural Network (CNN) architecture used in the image-based method. The input of CNN is a binary scale landslide image, and the output of CNN is the probability of a landslide image belonging to an earthquake or rainfall-induced class.





## 4.1 Evaluation of the first method (geometric features based classification)

Combining all the landslide inventories with known triggers lead to $26,501$ samples ($n_{total}$), out of which $16,196$ are earthquake-triggered landslides ($n_{earthquake}$) and $10,305$ are rainfall-induced landslides ($n_{rainfall}$). As the number of earthquake-triggered landslides is much larger than the number of rainfall-induced landslides, we use equal numbers of each trigger class to avoid any class-imbalance problems. To avoid selection bias and overfitting, we apply 10-fold cross-validation. $k$-fold cross-validation splits the combined classes dataset into $k$ random subsets where each iteration of cross-validation, $k-1$ folds are used for training and the remaining fold for testing. We use $20,610$ samples ($n_{rainfall} = n_{earthquake} = 10,305$) for cross-validation and to get generalizable results we employ 1000 runs of cross-validation. In each run of cross-validation we randomly select $10,305$ earthquake samples from $16,196$ earthquake landslides. We achieved $86.15 \pm 0.22\%$ classification accuracy for earthquake, $85.29 \pm 0.19\%$ for rainfall, and $85.73 \pm 0.16\%$ as the mean classification accuracy.

For the second split configuration, we trained the random forest classifier on five inventories and tested it on the sixth inventory. For earthquake triggered inventories the method achieved classification accuracy of $66.62 \pm 0.65\%$, $75.59 \pm 0.34\%$ and $85.22 \pm 0.20\%$ for the Hokkaido ($n_{train} = 20,610, n_{test} = 3,256$), Iwata ($n_{train} = 20,610, n_{test} = 4,160$) and Niigata ($n_{train} = 14,832, n_{test} = 8,780$) inventories (for geographical locations of these inventories see Figure 1). For rainfall induced inventories, we achieved classification accuracy of $83.63 \pm 0.41\%$, $69.40 \pm 0.61\%$ and $92.12 \pm 0.25\%$ for Kumamoto ($n_{train} = 9,482, n_{test} = 5,564$), Fukuoka($n_{train} = 16,762, n_{test} = 1,924$) and Saka ($n_{train} = 14,946, n_{test} = 2,817$) region. In each one of the the case we took equal number of earthquake and rainfall triggered landslide samples to avoid any class imbalance issues ($n_{earthquake} = n_{rainfall}$). The low standard deviation in classification accuracy shows that results are stable with change in training samples.

## 4.2 Evaluation of the second method (topological features based classification)

In the first test and training set split configuration, as in Section 4.1, we used $n_{total}= 20,610$ (total number of samples), $n_{earthquake}$=$10,305$ (number of earthquake-triggered samples) and $n_{rainfall}$ =$10,305$ (number of rainfall-induced samples), keeping numbers of each trigger class equal to avoid class imbalance. We first identify the top ten relevant topological features out of thirty features, employing 1000 runs of 10-fold cross-validation of random forest. Using these top ten relevant topological features as the feature space for the random forest classifier, we carry out 1000 runs of 10-fold cross-validation to get generalized classification accuracy. The method achieved above $94\%$ classification accuracy for earthquake, rainfall, and mean class classification.

In the second split configuration, this method achieves above $90\%$ accuracy for the Iwata, Niigata, Kumamoto, and Saka inventories. For the Hokkaido and Fukuoka region, the method achieves above $80\%$ classification accuracy (see Figure 7). The number of training and testing samples for each case is the same as in Section 4.1.





## 4.3 Evaluation of the third method (image based classification)

As explained above in section 3.3 we removed large landslides from the analysis leading to $n_{total} = 24,311$, $n_{earthquake} =$

235  $14,892$, and $n_{rainfall} = 9,419$. We used an equal number of training samples of the coseismic and rainfall-induced landslides to avoid any class imbalance issues. We used 100 runs of different test and training sets instead of different runs of 10 fold cross-validation as convolutional neural networks are computationally expensive. The method achieved above $85\%$ classification accuracy for earthquake, rainfall, and mean class classification.

For the second split configuration, the method achieved above $80\%$ accuracy for the Saka region ($n_{train} = 13,738, n_{test} =$

240  $2,550$). For the Niigata ($n_{train} = 12,780, n_{test} = 8,502$), Kumamoto ($n_{train} = 8,276, n_{test} = 5,281$) and Fukuoka ($n_{train} = 15,662, n_{test} = 1,588$) region the method achieves accuracy of above $70\%$. The Method achieves $67\%$ accuracy for the Hokkaido inventory ($n_{train} = 18,838, n_{test} = 2,431$). In each one of the the cases, we took equal number of earthquake

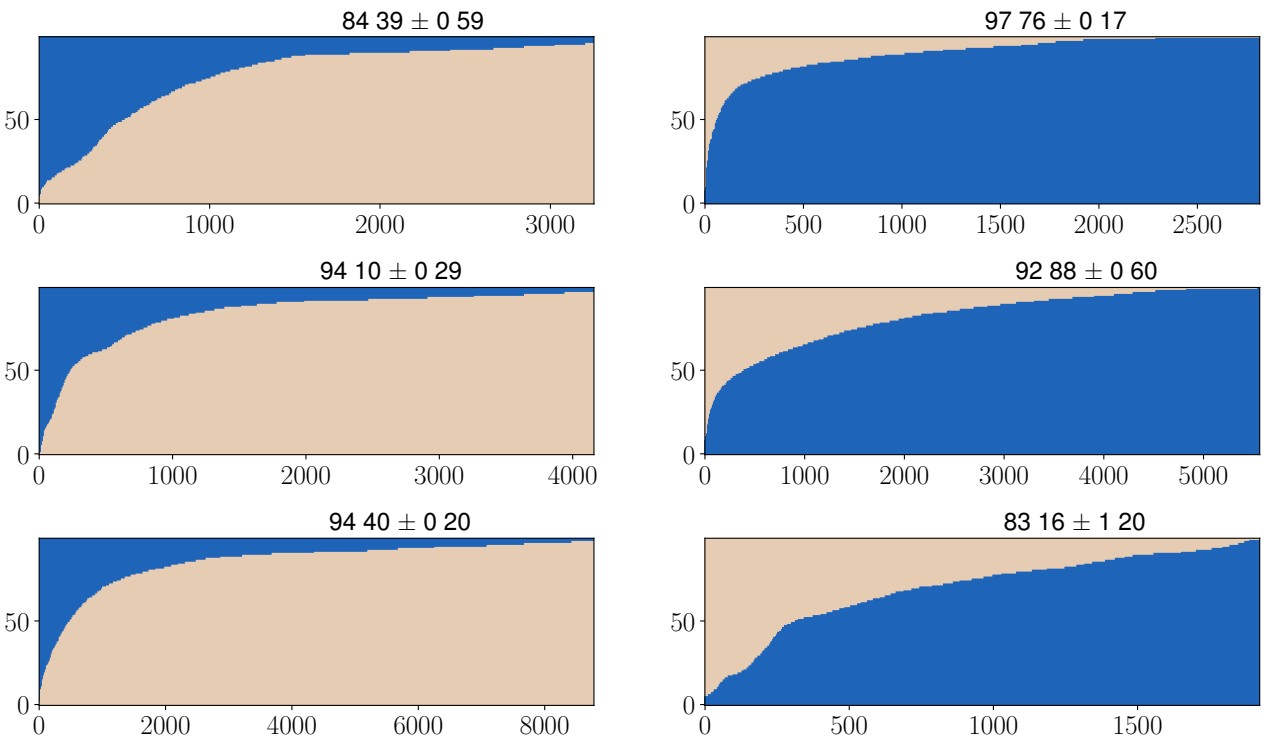

**Figure 7.** The topological features-based method (second method) accuracies for all the six known triggered inventories. The model is trained on five inventories in each case and tested on the sixth inventory. The y-axis in the plot shows the probability of landslides belonging to the earthquake and rainfall class, and the x-axis shows the sample index of landslides.





and rainfall induced landslide samples to avoid class imbalance issues.

## 5 `Landsifier` library

One of the main aims of this paper is to introduce `Landsifier`, a Python library we built to provide the landslide research community with a user-friendly computational package to implement the methods described above. At the moment, we have made the code available on the corresponding author's GitHub: `https://github.com/kamalrana7843/Landsifier.git`. Furthermore, we plan to publish `Landsifier` library under an open-source license after the acceptance of this manuscript for publication. In Appendix B we provide details of the library and brief descriptions of the available functionalities.

## 6 Discussion

The geometric properties of landslides can provide information about their trigger (Taylor et al., 2018). Our preliminary work on landslide trigger classification demonstrated that landslides with identical triggers share similar geometric properties, which could be exploited to classify landslide triggers—see the publication Rana et al. (2021) and briefly reproduced results here in sections 3.1, 4.1 and Appendix B1. In this work, we further expanded our initial approach by adding two additional methods for landslide triggers classification and a Python library `Landsifier` to implement them. One of these two new methods uses 3D shapes of landslides for their trigger classification by incorporating the elevation information. We compute topological features of these 3D shapes using Topological Data Analysis (TDA) and use the features as an input to a machine learning-based algorithm—random forest. The other method uses binary scale landslide polygon images as an input to Convolutional Neural Networks (CNN) for the classification. Using six landslide inventories spread over the Japanese archipelago, we showed that each method exhibit strong performance to classify landslide triggers. However, each method has its strengths and limitations that primarily depend on training and testing landslide data quality, quantity, and location. We explained each methods' strengths and limitations in different conditions in this section. Before providing some hints about potential future work and opportunities that could arise from using `Landsifier` library, here, we also present and discuss the results of each three method on the seventh Kumamoto unspecified inventory.

The landslide data quality depends on the data acquiring technique; e.g., landslide data obtained using aerial or satellite images are much higher quality than the data acquired via field campaigns. Geologists collect landslide data acquired via field campaigns, and by nature, such inventories tend to fail to represent the smaller landslides and cover the larger landslides (Ozturk et al., 2020). Whereas landslides inventories acquired via aerial or satellite images cover both small and larger landslides and are called complete inventories as they adequately capture landslides of various sizes in their respective study area, e.g., see (Schmitt et al., 2018). Training the geometric feature-based and image-based methods on landslide planforms with landslide data acquired via satellite or aerial images and testing on data acquired via field campaign or vice-versa could lead to



| Inventory Region | Geometric features based method (%) | Topological features based method (%) | Image based method (%) |
|---|---|---|---|
| Hokkaido | 67 | **84** | 68 |
| Iwata | 76 | **94** | 67 |
| Niigata | 85 | **94** | 77 |
| Saka | 92 | **98** | 88 |
| Kumamoto | 84 | **92** | 78 |
| Fukuoka | 69 | **83** | 70 |

**Table 1.** The table shows landslide classification results using the three methods. The model is evaluated on all possible training set combinations of the five inventories and tested on the sixth inventory.

biases in landslide classification results. The methods based on landslide planforms shape consider the area and perimeter as
the most important features and rely on the information that coseismic landslides are generally larger than rainfall-induced
landslides (Rana et al., 2021) (e.g., Taylor et al., 2018; Tanyaş et al., 2021). So, a testing inventory triggered by rainfall but
lacks smaller landslides due to field campaign acquisition technique could be classified as earthquake-triggered—given that
training inventories are satellite or aerial image-based. We recommend using similar field campaign acquired inventories with
known triggers to train the models for more accurate classification in such a scenario. Another option is to sample landslides
from the satellite or aerial image-based inventories that resemble the size distribution of the testing data acquired via field
campaign. This shortcoming motivated us to offer another alternative solution relying on topological analysis of 3D shapes of
landslides.

Landslides are 3D shapes; thus, using 3D shapes of landslides instead of 2D could provide additional information related to
the landslide morphology. Consequently, a 3D landslide shape-based method might elevate classification accuracy, especially
in regions without proper training and testing data of similar quality. We use TDA, a method rooted in algebraic topology,
to compute topological features of a landslides' 3D shapes to classify landslide triggers. In Table 1 one can observe that
the TDA-based method indeed performs better than the other two methods. However, TDA-based measures encode landslide
morphology; hence, if testing and training inventories share similarities in the geomorphology of the studied regions (spatial
autocorrelation) (Oksanen and Sarjakoski, 2005), then the trigger prediction is highly influenced by training inventory. Geomet-
ric features and image-based methods are less sensitive to the geomorphological similarities between the training and testing
landslide inventories, as these only use the 2D landslide planforms. Although the image-based performs satisfactorily only
when adequate large training data is available. Hence, we recommend using geometric or topological features-based methods
in inventories with limited landslide counts.

We applied each method to classify landslides triggers in the Kumamoto unspecified inventory having an undocumented
trigger to demonstrate the real-world application of the `Landsifier` library. Out of 612 landslides in the inventory, the ge-
ometric feature-based and topological feature-based classified 604 and 612 landslides as earthquake-triggered. In comparison,





the image-based method uses 164 landslides after removing landslides having width and length greater than 180 meters (see section 3.3 for more details) and classified all of the landslides as seismically triggered. As each method classifies the majority of the landslides as earthquake-triggered, we are confident that earthquake is the most likely trigger for most of the landslides

in this inventory. Moreover, the Kumamoto unspecified inventory documents landslides along the rims of the Aso Caldera, the active volcano Mount Aso shakes the surrounding area frequently triggering landslides within its vicinity (Saito et al., 2018). Hence, it is very likely that this inventory is consistent of landslides of cosesimic origin.

Considering the above discussions, in future work, we plan to explore further the sensitivity of our trigger classification methods to spatial autocorrelations. We will also examine the influences of landslide size distributions on each method. More-

over, we will consider model transferability to different regions by extensively testing these methods on national landslide inventories, e.g., India, Nepal, Taiwan, and the USA. Our methods could also provide other opportunities. For example, assessing landslide-prone regions as an alternative to landslide susceptibility measure using TDA. Also, TDA could be used to classify landslide types, according to the types described in Cruden and Varnes (1996) and Varnes (1978). We plan to further develop the current version of the `Landsifier` by incorporating a landslide type classifier in the next version. This method

will be able to find the analogy between an observed landslide and a generic landslide types by Cruden and Varnes (1996).

## 7  Conclusions

The landslide triggering mechanism is crucial information to develop landslide hazard models, e.g., a landslide hazard model for extreme rainfall incidents requires landslide inventories related to rainfall events only. However, modern automated landslide mappers for continuous monitoring and historical landslide inventories rarely report the landslide triggering mechanism.

Missing triggers in the landslide inventories decrease their efficacy for landslide hazard models. In this work, we developed a Python library, `Landsifier`, containing three methods for landslide trigger classification by exploiting landslide planforms and 3D shapes. To develop the first two of these methods, we combined geometric and topological features with machine learning, and in the third method, we used deep learning. The latter two methods are new, i.e., we are reporting them here for the first time.

We use seven landslide inventories spread over the Japanese archipelago. Six among these seven inventories have known triggers, while the seventh inventory has a missing trigger. We applied each method to all possible sets of five training inventories and one testing inventory using six known triggered inventories. Moreover, we took different training and testing sets of landslides by mixing all known triggered landslides inventories following the k-fold cross-validation. The achieved results demonstrate that the methods are robust and capable of classifying triggers of landslide inventories with high accuracy (70%–

95%). To demonstrate the real-world application of our toolbox, we also applied the three methods to the seventh inventory without any triggering information and classified it as an earthquake-triggered inventory.

Python based `Landsifier` library provides a user-friendly computational package to implement the methods described above to the landslide research community. We anticipate that the landslide research community will find the `Landsifier` library helpful in finding the trigger mechanism of inventories or individual landslides. The presented methods and the library





could be deployed in any region of the world with adequate training data from areas with similar climatic and tectonic features. Furthermore, our tools are easy to use as they require only shapefiles of landslide polygons as input. At the moment, we have made the code available on the corresponding author's GitHub: `https://github.com/kamalrana7843/Landsifier.git`. Furthermore, we plan to publish `Landsifier` library under an open-source license after the acceptance of this manuscript for publication. In Appendix B we provide details of the library and brief descriptions of the available

functionalities.

## Appendix A: Random forest

Random forest (RF) is a decision-tree based ensemble-learning method, a proven and powerful technique for classification and regression (Barnett et al., 2019; Biau, 2012; Biau and Scornet, 2016; Breiman, 2001; Kursa, 2014; Chaudhary et al., 2016; Rodriguez-Galiano et al., 2012). The random forest classifier consists of multiple classifiers, where each classifier bootstraps

the training data samples (Breiman, 2001; Liaw et al., 2002). Bootstrapping in each random forest classifier is done by selecting $N$ samples randomly from training samples of size $N$ with replacements. For $N$ training samples bootstrapping $N$ times leads to the approximate selection of $2/3$ of training samples (Azar et al., 2014; Belgiu and Drăguţ, 2016). Hence, each tree in a random forest classifier is trained independently using around $2/3$ of the training samples selected using bootstrapping.

In a binary classifier, as in our case, each parent node $q$ splits into two daughter nodes: right $r$ and left $l$. Instead of selecting

all the $p$ features for node split, a subset of features $m$ ($m=\sqrt{p}$) is selected randomly for each node split (Azar et al., 2014; Okun and Priisalu, 2007). Among $m$ features, one of the features selected for the node split is based on optimizing a criterion. The criterion is called the 'Gini Index,' which measures the features' impurity to the classes. The Gini index of right $r$ and left $l$ daughter nodes are calculated as:

$$G_r = 1 - P_{r1}^2 - P_{r2}^2 \tag{A1}$$

$$G_l = 1 - P_{l1}^2 - P_{l2}^2, \tag{A2}$$

where $P_{rj}$ ($P_{lj}$) is the probability of samples in the right (left) daughter nodes having class $j$. The Gini index is calculated for each predictor in the subset of predictors $m$, and the features that maximize the change in Gini index is chosen for node split. Change in Gini-index is calculated as:

$$\Delta\theta(s_q) = G_q - \rho_{rq}G_r - \rho_{lq}G_l, \tag{A3}$$

where $\rho_{rq}$ ($\rho_{lq}$) are the ratio of the number of data points in daughter nodes $r$ ($l$) to the total number of points in the parent node $q$ (Kuhn et al., 2013; Zhang and Ma, 2012). The process of splitting nodes continues until a stopping criterion is met, e.g., when no further samples are remaining, or the Gini-index of parent nodes is lower than the daughter nodes.

The steps for constructing trees in the random forest are as follows:

(i) Select bootstrap samples of size $N$ from training samples of size $N$.




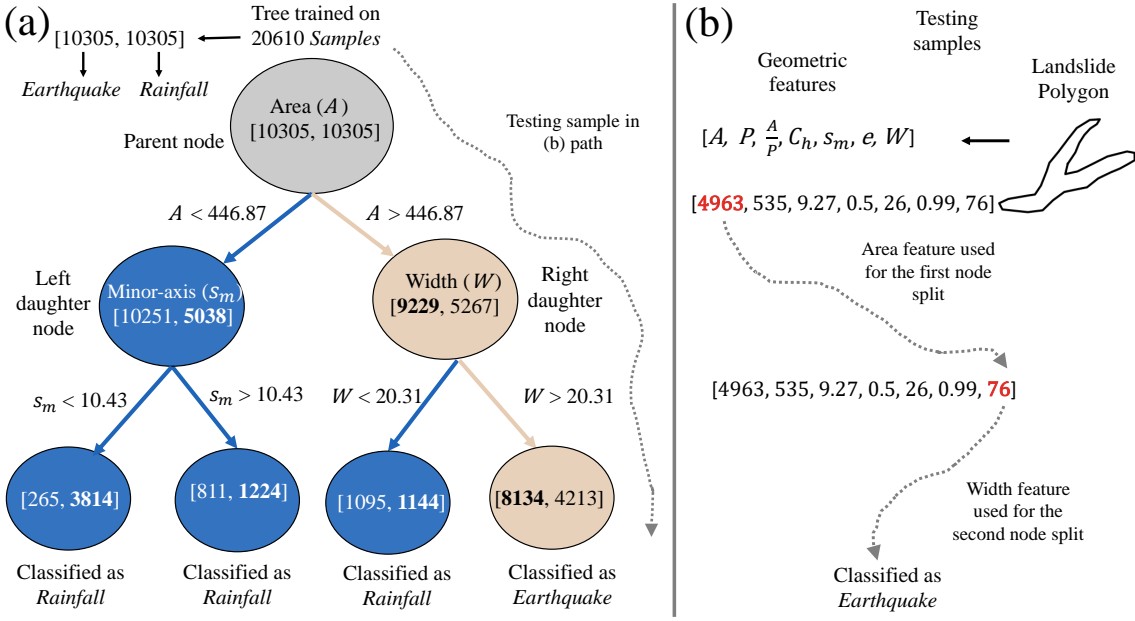

**Figure A1.** (a) The sample architecture of one of the trees of random forest classifier. The tree is trained on 20610 landslides samples with 10305 each earthquake and rainfall trigger class. Feature vector ($[A, P, C_h, W, s_m, \frac{A}{P}, e]$) represents landslide geometric property corresponding to each landslide sample. For illustration purposes, the tree is grown to only depth three. (b) testing sample of landslide tested on the tree shown in (a). The sample landslide polygon is classified as an earthquake.

(ii) Randomly select $m$ variables among $p$ variables for the node split.

     (iii) Choose one variable among $m$ variables that best split the node according to the Gini-index criterion.

     (iv) Continue repeating steps (i) to (iii) until the stopping criterion is met.

For testing, each tree classifier predicts the class of testing sample independently, and the class with majority votes is the final outcome of random forest (Kuhn and Johnson, 2013; Pal, 2005; Arabameri et al., 2021; Belgiu and Drăguţ, 2016).

In random forest, bootstrapping training samples selection and random selection of features for a node split reduces the correlation between trees. This technique has proven to improve the predictive power of ensemble learning (Azar et al., 2014). In addition, random forest assigns each feature a score that provides its relative importance (Qi, 2012; Friedman et al., 2001). Features with low relative scores should be discarded as they are neutral to the model accuracy and increase the model complexity.



## Appendix B: Details of `Landsifier` library

`Landsifier` is a Python library we built with version 3.6 of Python and the code is available on GitHub: `https://github.com/kamalrana7843/Landsifier.git` (we will publish this library under an open-source license after this manuscript is accepted for publication). On this link, prospective users can also find the list of Python packages used in the library. `Landsifier` contains three methods for landslide trigger classification, and these methods only use shapefiles from landslide inventories (two of these methods use 2D polygon shapes of landslides, while the third method uses the 3D shapes of landslides). This section describes various functions provided in the `Landsifier` library to implement the above methods and Figure B1 summarizes these functions in form of a flowchart. Also, Figure B2 shows a sample output of the `Landsifier`.

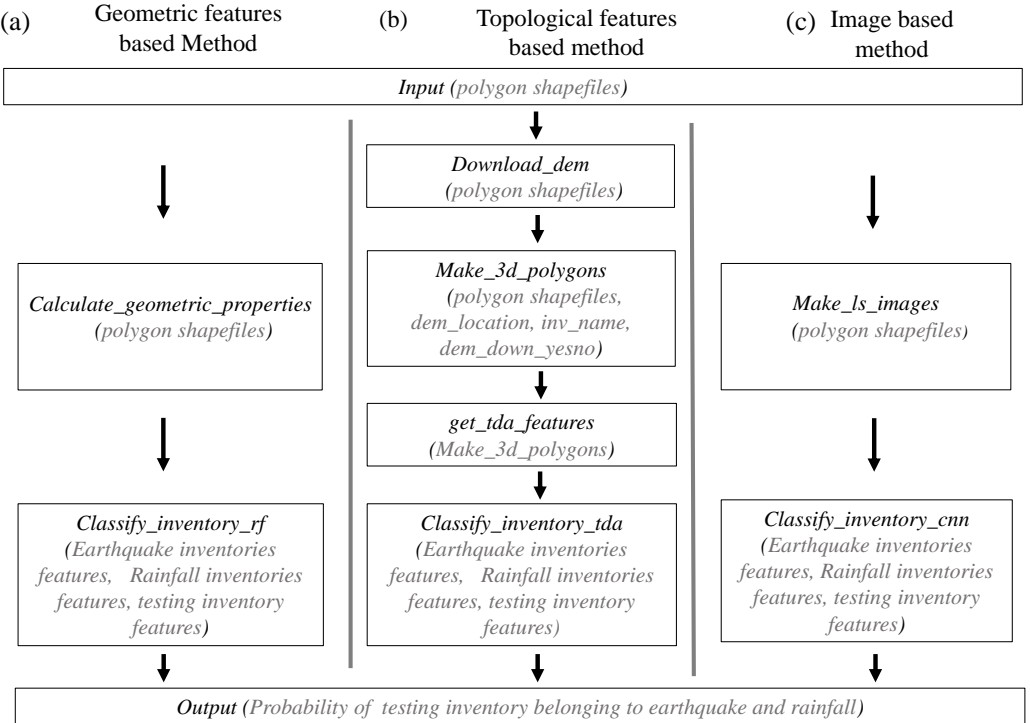

**Figure B1.** The figure shows the flowchart of implementations of all the three methods using functions and their variables used in the `landsifier` library. All three models use polygon shapefiles as an input to the model and provide the probability of landslide belonging to earthquake and rainfall as an output (a) geometric features based method (b) topological features based method (c) image-based method.

## B1 Functions for geometric features based classification

Below we list functions to implement the geometric features-based classification, details of the method can be found above in section 3.1 and in our publication (Rana et al., 2021). Note below we have described functions in a form that this method can

Natural Hazards
and Earth System
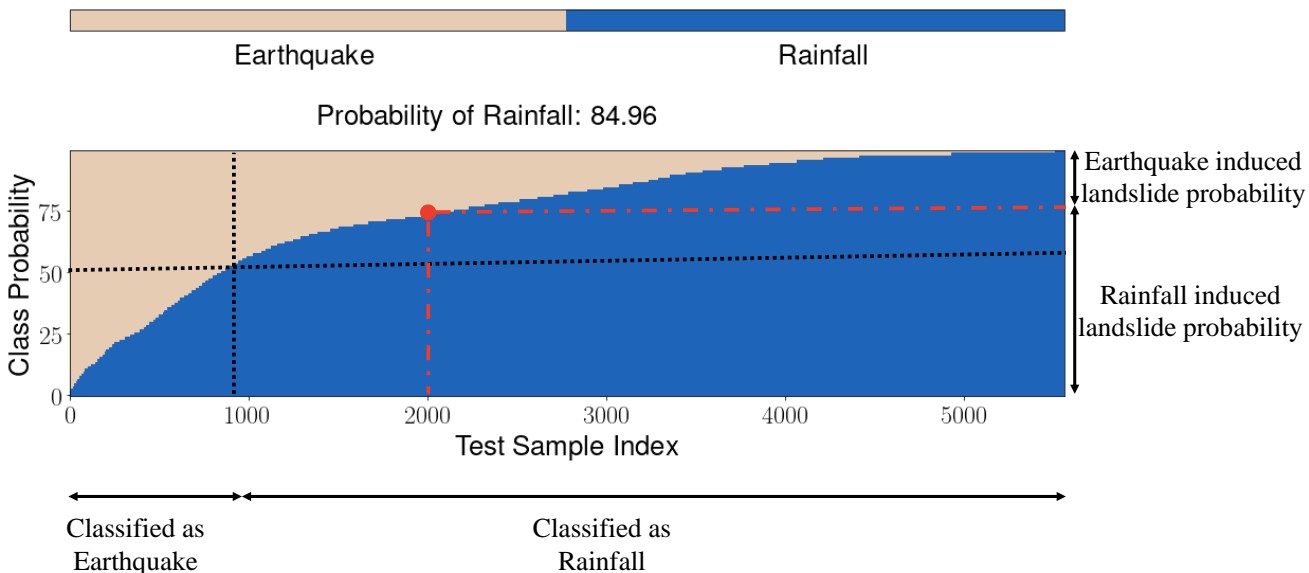

**Figure B2.** The output of the geometric feature-based method when Kumamoto inventory is testing inventory and the rest five inventories are used as training inventories. Each method in the `landsifier` will produce similar outputs. The y-axis in the plot shows the probability of landslides belonging to the earthquake and rainfall class, and the x-axis shows the sample index of landslides. For each landslide in the testing inventory, all the models give a probability of landslides belonging to earthquake and rainfall-induced classes. The predicted trigger of most of the testing landslides is the probable trigger of the testing inventory.

be used for inventories with unspecified triggers, i.e., unknown ground truth.

`latlon_to_eastnorth(latlon_polydata)`: This function takes polygons data in $(longitude, latitude)$ coordinates as an input and provides polygon data in $(easting, northing)$ coordinates as output to the function. This function is used to
get landslide polygons in $(easting, northing)$ coordinates when polygon data in shapely files are in $(longitude, latitude)$ coordinates.

`Calculate_geometric_properties (polygon_shapefile)`: As the name suggests, this function calculates the geometric properties of each of the landslide polygons present in shapefile. This function takes polygons shapefiles
`(polygon_shapefile)` as input, converts polygon data into $(easting, northing)$ coordinates if required using the
`latlon_to_eastnorth` function, and then provides the geometric properties of polygons as output to the function. For each landslide polygon it calculates a vector($[A, P, C_h, W, s_m, \frac{A}{P}, e]$) containing polygon geometric properties as output to the



function. All the geometric properties of the landslide polygon are calculated using the *shapely* package in Python.

`classify_inventory_rf (earthquake_inventory_features, rainfall_inventory_features, test_inventory_features)`: This function takes the earthquake-triggered inventories ($earthquake\_inventory\_features$), rainfall-induced inventories ($rainfall\_inventory\_features$) and testing inventories ($test\_inventory\_features$) geometric features as the input. Within the function, it trains the random forest algorithm on training data containing equal samples of the earthquake and rainfall-induced class. The output of the function is the probability of testing landslides belonging to each

trigger class.

### B2  Functions for topological features based classification

Below we list functions to implement the topological features-based classification, details of the method can be found above in section 3.2. Note below we have described functions in a form that this method can be used for inventories with unspecified triggers, i.e., unknown ground truth.


`download_dem (polygon_shapefile)`: This function takes shapefile of landslide polygons as an input and downloads the *Shuttle Radar Topography Mission* digital elevation model (DEM) of resolution 30 meters corresponding to inventory region (Farr et al., 2007). It takes the bounding box over the entire inventory location and calculates the (`minimum latitude, minimum  longitude`) and (`maximum latitude, maximum longitude`). Using the elevation

package in Python, it downloads the DEM data of a region bounded by `minimum latitude, minimum longitude, maximum latitude,` and `maximum longitude` coordinates. This function downloads all the tiles (one tile constitutes $1° \times 1°$ region in both latitude and longitude) of the inventory region and combines all the tiles into one file corresponding to one inventory.

`make_3d_polygons (polygon_shapefile, dem_location, inv_name, dem_down_yesno )`: This function takes landslide polygon shapefiles (`polygon_shapefile`), DEM path location (`dem_location`), inventory name (`inv_name`) and Boolean parameter (`dem_down_yesno`) as input and provides 3D point cloud data of landslides as output. This function carries out several tasks. First, it downloads the DEM data corresponding to the whole inventory region in path location (`dem_location`) with inventory name (`inv_name`) using `download_dem` function if `dem_down_yesno` is True.

If users already have DEM corresponding to inventory in path location (`dem_location`) with inventory name (`inv_name`) then (`dem_down_yesno`) is False. Then corresponding to each landslide polygon it interpolates the DEM data around the bounding box of the polygon. Using the shapely package, the function removes all the interpolated data outside the outline of the landslide polygon and takes elevation data only within the landslide.

`get_tda_features (three_d_data)`: This function takes the 3D shape of landslides point cloud data (`three_d_data`) as an input and provides machine learning features corresponding to each 3D landslides as an output to function. This func-





tion calculates the persistence diagram using Vietoris Rips persistence for each 3D landslide, and then using the persistence diagram, it calculates the following TDA metrics: persistence entropy, average lifetime, number of points, betti curve based measure, persistence landscape curve based measure, Wasserstein amplitude, Bottleneck amplitude, Heat kernel-based mea-
sure, and landscape image-based measure corresponding to each homology dimension–0, 1, and 2. These TDA metrics are used as a feature space for the machine learning algorithms.

`classify_inventory_tda` (`earthquake_inventory_features`, `rainfall_inventory_features`, `test_inventory_features`): This function takes training earthquake inventory's (`earthquake_inventory_features`),
training rainfall inventory's (`rainfall_topological_features`) and testing inventory's (`test_inventory_features`) TDA based features as input to function. Inside the function, it first selects the top 10 features with the highest feature importance using training data. It then combines an equal number of training earthquake and rainfall samples to avoid any class imbalance problem. It trains the random forest algorithm on training data and predicts the probability of testing landslides belonging to each trigger class.


**B3    Functions for image based classification**

Below we list functions to implement the image-based classification, details of the method can be found above in section 3.3. Note below we have described functions in a form that this method can be used for inventories with unspecified triggers, i.e., unknown ground truth.


`increase_resolution_polygon` (`poly_data`): This function takes a single polygon coordinates data (`poly_data`) in $(easting, northing)$ coordinates as input and increases the number of points between any two adjacent vertexes of the polygon within the function. This function is useful in creating smooth binary scale landslide polygon images. The output of the function is landslide polygons coordinates data having multiple points between the adjacent vertex of polygons.


`make_ls_images` (`polygon_shapefile`): This function takes polygon shapefiles (`polygon_shapefile`) as an input and provides landslide polygon images as an output to the function. It creates $N \times N$ ($N = 64$ in our case) pixel image with binary values of 0 or 255 for each pixel. This function first increase the number of data points in polygons using `increase_resolution_polygon` function and then takes a bounding box of polygon and transforms the coordinates of polygons by subtracting polygon $(minimum\_easting, minimum\_northing)$ value from each point in the polygon. Then divide each point in polygon $(easting, northing)$ value by resolution of pixels (desired spatial distance between any two adjacent horizontally or vertically pixels) and convert them into nearest integers. Then for each pixel $(x, y)$ the value of the pixel is 255 if there exists a point in the polygon with coordinates $(x, y)$ otherwise the value of the pixel is 0. This function also removes those landslides having length and width of bounding box greater than 180 meters as the image of a polygon has some





restrictions on maximum landslide polygon it can have (resolution of pixels ( 3 meters) × N = 192 meters).

`train_augment (train_data, train_label)`: This function takes input training landslides data (`train_data`) and training labels (`train_label`) as input. The main idea behind using `train_augment` function is to augment the training data as CNN is data extensive algorithm. It rotates each image by 90°, 180°, 270°and flip the image vertically and

horizontally to increase the number of training samples. The output of the function is augmented training data and labels.

`classify_inventory_cnn (earthquake_inventory_images, rainfall_inventory_images,`
`test_inventory_images)`: This function takes training earthquake inventory images (`earthquake_inventory_images`), training rainfall inventory images (`rainfall_inventory_images`) and testing inventory images (`test_inventory_images`)

as input to the function. Within the function, it combines an equal number of training earthquake and rainfall samples to avoid any class imbalance problem and then augments the training data by using the `train_augment` function. Then it trains the CNN algorithm on augmented training data and predicts the probability of testing landslides belonging to each of the trigger classes.

*Code availability.* The source code and future updates are available in the GitHub repository ( `https://github.com/kamalrana7843/` `Landsifier.git`).

*Data availability.* The landslide inventories used in this paper are publicly available by Geospatial Information Authority (GSI) and the National Research Institute for Earth Science and Disaster Resilience (NIED). The 30 meters SRTM DEM data used is also publicly available by NASA and downloadable via (`https://www2.jpl.nasa.gov/srtm/`).

*Author contributions.* All authors contributed to the writing and reviewing of the manuscript. KR developed the code. NM and UO interpreted the results and supervised the work.

*Competing interests.* The authors declare that they have no conflict of interest.

*Acknowledgements.* This project is supported by Co-PREPARE project (No: 57553291) by German Academic Exchange Service (DAAD). KR acknowledges support from RIT's Steven M. Wear Endowed graduate fellowship and NM acknowledges support through RIT's FEAD
grant. UO acknowledges funding from the Research Focus Point "Earth and Environmental Systems" of the University of Potsdam.



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
