# Peer review of "Landsifier v1.0: a Python library to estimate likely triggers of mapped landslides"

_Natural Hazards and Earth System Sciences, 2022_

## Author Comment (AC1)

**Response to reviewer 1**

September 13, 2022

We thank the reviewer for acknowledging our work, taking the time to read it, and providing suggestions to improve the work. We hope the reviewer will find our revised paper better suited for publication. We have highlighted changes to the paper in response to the reviewer's comments in red in the revised article and in response to the reviewer's letter.

**Abstract**

*Well written and rather clear. Perhaps should be stressed better the differences among three machine learning methods.It could be useful to follow the same order of presentation that has been illustrated in the introduction (RDF,CNN,TDA).*

We appreciate the reviewer comment and have incorporated the suggested changes. Now, the order of the presentation in abstract is same as in the Introduction (RDF, CNN, TDA). We have also included 2 new sentences explaining the basic differences between the methods.

**1 Introduction**

*Very good the stress about the difference among earth-triggered and rainfall-triggered events. It is a common problem also today since there is not a standard way to classify landslide information. It should be better stressed that the whole work has been included in a new python library and that it is one of the unique in this field.?*

We thank the reviewer for acknowledging our work and appreciate the reviewer's comments about the Introduction section. We have included reviewer suggestions about explaining the uniqueness of the whole work presented in a new python library. Please see the red text on page 3 between lines 60 and 65 in the revised paper. Also, we reproduce the updated text below for the reviewer's convenience.

"Landsifier is the first-ever library built for estimating likely triggers of mapped landslides, the methods used in this library to find landslides' triggers are new. Two of these methods are introduced in this paper for the first time, while the third was published in our preliminary work (Rana et al., 2021)."

**2 Data**

*Ok, all information required and necessary for the case study are listed correctly!*

We appreciate the reviewer comments about the Data section.

**3 Methods**

**3.1**

*ok well clear. Perhaps, a brief explanation on the differences between earthquake and rainfall triggered shapes should be included even though the reader miss the key information about the classificator.*

We have added a paragraph highlighting the geometric dissimilarities between the two groups. They are marked in red on page 5 in section 3.1 of the revised paper. We reproduce the new paragraph below.

"In Rana et al. (2021), we analyzed the distributions of geometric properties of the earthquake and rainfall polygons and found geometric dissimilarities between earthquake and rainfall polygons' shapes. Earthquakes polygons are more likely to have a compact shape (as measured by convex hull-based measure) than rainfall polygons. Moreover, earthquake polygons have more chances to have a larger area $(A)$, perimeter $(P)$, the ratio of the area to the perimeter $(\frac{A}{P})$, and minimum width $(W)$ than rainfall polygons. In contrast, rainfall polygons have a larger eccentricity $(e)$ than earthquake polygons of an ellipse fitted to the polygon. Rainfall polygons are more sinuous in shape leading to the smaller minor axis and larger major axis leading to the larger eccentricity of the ellipse fitted to the polygon (Rana et al., 2021)."

**3.2**

*the method is presented clearly but in my opinion is missed a point: why we need all these infromation extracted by the DEM? It should be remarked at least to highlight you are presenting this second approach*

We have incorporated the above remarks into the paper. We added these new features to improve the model accuracy, and they did indeed increase the final model's classification accuracy. Please see the red text in the revised paper's first paragraph on page 5, reproduced below.

"The topological properties of the landslide's 3D shape extracted using DEM provide additional insights into the landslide triggers, which might further improve the accuracy of the landslide trigger classification."

**3.3**

*3.3 here is clear that the output is a probability to belong to one or to the other class. And the presentation of the CNN method in more readable!*

We appreciate the reviewer comment

**4 Landsifier model evaluation**

*from 198-202 it is not clear how the simulation was carried out. Please revise it and may be a short table could be helpful in this sense*

We have now revised and added more information. Please see the text highlighted in red under section 4 in the revised paper. For the reviewer's convenience, we have reproduced the new lines below.

"We used two different testing configurations to evaluate the efficacy of our methods. Finding the triggers of individual landslides irrespective of their inventories is the first testing configuration. Here, we combined all the known trigger landslides from all six known triggered inventories and then split the combined landslides data into various training and testing sets following the $k$-fold cross-validation framework. In this testing configuration, landslides in each training and testing set are from all six landslide inventories. The second testing configuration finds the trigger of landslide inventories itself. We used all the possible combinations to train the algorithm on five known trigger inventories and test it on the sixth inventory. In this second testing configuration, landslides in the testing set are from a single inventory. Note that there are seven inventories in the analyzed data set, and six have known triggers. We present the analysis of this seventh inventory (Kumamoto unspecified) with unknown triggers in section 6."

**4.1**

*OK well presented and clear*

We appreciate the reviewer feedback.

**4.2**

*the same of 4.1*

We appreciate the reviewer feedback.

**4.3**

*the same of 4.1*

Thank you.

**5 landsifier library**

*Ok for the riminder but since it represents the TITLE of the work probably few words should be spent here. For example: which functions are embedded in? Settings and options? Is it fast or slow in computation? Just some characteristics that can involve the reader to download and test it.*

In response to this comment, we have added new text in several places in section 5. We have briefly explained the functions, options, and computational speed of the methods included in the library. Please see the red-colored text on pages 14-15. Also, we have reproduced below the new text added to the paper in response to this comment.

"Apart from three different methods for landslide trigger classification, the library also contains other useful functions like calculating geometric properties of landslide polygons, converting polygons to binary scale images, downloading DEM corresponding to an inventory region, and converting 2D landslide polygon to 3D landslide shape (see Figure 3a and 5a). Please refer to Appendix B for further details about the library functions (Figure B1). Each of the three methods used in the library is simple to use and only requires polygon shapefiles as input. Also, the computation process is relatively fast; for example, the geometric, image, and topological features-based method takes less than 5, 15, and 45 minutes for training on 20,000 landslides (equal earthquake and rainfall samples) on a windows machine with 16 GB of RAM (Random-Access Memory) using only landslide shapefiles as input. Moreover, none of the methods requires a GPU (Graphics Processing Unit)".

**6 Discussion**

*The discussion are well written but in my opinion should be rather organized in order to highlight better the outcomes of the study.*

We have updated the discussion section in the revised version of the paper. Please see the red-colored text in the discussion (section 6). See red-colored text in the discussion (under section 6).

Which is the best technique adopted? Cost e-benefit of each technique? Computational demand? Accuracy? All of the questions are expected by the reader aftere the presentation of the new python library.

Each technique has its strengths and limitations that primarily depend on training and testing landslide data quality, quantity, and location.

**Data Quality:** Training the geometric feature-based and image-based methods on landslide planforms with landslide data acquired via satellite or aerial images and testing on data acquired via field campaign or vice-versa could lead to biases in landslide classification results. Landslide data acquired via field campaigns tend to fail to represent the smaller landslides and cover only the larger landslides (Ozturk et al., 2020). Whereas landslides inventories acquired via aerial or satellite images cover both small and larger landslides. The methods based on landslide planforms shape consider the area and perimeter as the most important features and rely on the information that coseismic landslides are generally larger than rainfall-induced landslides. In such scenerio we recommend to use topological feature based method as it also includes the morphology of the landslides region.

**Data location:** TDA-based measures encode landslide morphology; hence, if testing and training inventories share similarities in the geomorphology of the studied regions (spatial autocorrelation) (Oksanen and Sarjakoski, 2005), then the trigger prediction is highly influenced by training inventory. Geometric features and image-based methods are less sensitive to the geomorphological similarities between the training and testing landslide inventories, as these only use the 2D landslide planforms.

**Data quantity:** The image-based will performs satisfactorily classification results only when adequate large training data is available. Whereas, geometric and topological features based methods will work well even absence of large landslide known triggered dataset

**Computational demand**: Each method and function used in Landsifier library is computationally fast. The geometric, image, and topological features-based method takes less than 5, 15, and 45 minutes for training on 20,000 landslides (equal earthquake and rainfall samples) on a windows machine with 16 GB of RAM (Random-Access Memory) using only landslide shapefiles as input. Moreover, none of the methods requires a GPU (Graphics Processing Unit), and we successfully ran these methods on a windows machine with 16 GB of RAM (Random-Access Memory).

**Accuracy:** As explained above the accuracy of each of the method depend on data quality, quantity, and location. In this paper, we used six known triggered landslide inventories having good quality data spread over the Japanese archipelago. As expected topological features based method achieves highest classification accuracy (above 90%) for both testing schemes (shown in Table 1 in main paper).

from 267-282: this statement is OK but should be more integrated in the discussion of the current work: without good quality landslide data the performance

of classification techniques may be not sufficient.

We have included text in discussion section explaining that without good quality landslide data the performance of classificaction techniques may be not sufficient. For the reviewer's convenience, we have reproduced the new lines below.

Lines 300-305: The performance of developed methods depends on landslides data quality and without similar data quality in training and testing set the accuracy of classification techniques could be insufficient to conclude the trigger of landslide inventory and also might lead to biases.

from 282-293: ok so should be stressed better the TDA peculiarities with respect to the other

We have included text in discussion section explaining the TDA peculiarities with respect to the other.

Reproducing lines 316-319: "TDA based method extracts topological information along with geometric information of landslide shape. Whereas, geometric features based method and likely Image based method use only geometric information of the landslide shape for landslide classification. We expect TDA based method will provide best landslide trigger classification results."

from 294-310: ok but future outcomes and expected improvemente should be better highlighted!.

We have included text in discussion section explaining future outcomes and expected improvements in the Landsifier library. We have included the potential use of Landsifier library for classifing the trigger of the large landslides $(\text{Area} > 90,000\text{m}^2)$ in the discussion section as these landslides are the most dangerous landslides and effect huge area. Moreover, we discussed more about using Landsifier for landslide type classification. Landslide type information plays a crucial role in landslide risk assessment which is usually missed in landslide databases (Loche et al., 2022).

Reproducing lines 336-338: We will also examine the influences of landslide size distributions on each method. Specifically, we plan to classify the trigger of large landslides $(\text{Area} > 90,000\text{m}^2$ as they are the most dangerous landslides and effect huge area by training each method on large landslides training dataset.

Reproducing line 340-342: Landslide type information plays a crucial role in landslide risk assessment which is usually missed in landslide databases (Loche et al., 2022). We plan to further develop the current version of the Landsifier by incorporating a landslide type classifier in the next version (e.g., Amato et al. (2021)).

**7 Conclusions**

*They are well reassumed but in my opinion the Landsifier novelties and key new element should better shown.*

We thank the reviewer for acknowledging our work. In the conclusions section, we refrained from speculating further about the potential of the Landsifier library. However, following the reviewer's comment, we have added the Landsifier novelties, key new elements, and opportunities arising from it to the end of the conclusion section. We have added several sentences in the conclusion section, and we reproduce them below for the reviewer's convenience.

Lines 365-368: "Two of the three methods included in the library are new and introduced here for the first time, while the third method is published in our previous work. To best of our knowledge Landsifier is the first python tool developed for landslide triggers classification, and also such a tool does not exist in other programming languages."

Lines 370-373: "Landsifier library also contains useful functions like finding geometric properties of landslide polygons, downloading DEM corresponding to an inventory region, and converting landslide polygon to landslide 3D shape, these elements could be useful for the landslide research community."

In particular, are there any software application such as for landslide census or analysis at catchment scale?

To our best of knowledge there is no software application such as for landslide census or analysis at catchment scale. Having said that, there are several advancement in recent years estimating landslide populations following large landslide triggers, for example the following papers from (Marc et al., 2017, 2016):

Marc, O., Meunier, P., and Hovius, N.: Prediction of the area affected by earthquake-induced landsliding based on seismological parameters, Nat. Hazards Earth Syst. Sci., 17, 1159–1175, https://doi.org/10.5194/nhess-17-1159-2017, 2017.

Marc, O., Hovius, N., Meunier, P., Gorum, T., and Uchida, T., A seismologically consistent expression for the total area and volume of earthquake-triggered landsliding, J. Geophys. Res. Earth Surf., 121, 640– 663, https://doi.org/10.1002/2015JF003732, 2016.

Is it a tool useful for susceptibility mapping or also for Civil protection purpuose??

The topological features-based method in the Landsifier library could be useful for susceptibility mapping or civil protection purposes. We have included the

potential use of the Landsifier library briefly in the conclusion section. Below is the text that we added in the conclusion section regarding the potential use of the Landsifier library:

Lines 374-376: "Landsifier is a modular software, we hope the landslide community will further improve the offered tool and expand the available functions for new applications such as classifying landslide types, assessing landslide-prone regions, and other possible usage are listed in the discussion section."

**Appendix**

*Well organizaed and rather clear. Perhaps the scheme B1 of appendix B should be moved to the chapter 5 and then described briefly since represents the core of your work (Landsifier library)*

Adding scheme B1 of Appendix B to chapter 5 will increase the length of the main paper, which might decrease the lay reader's attention. Also, B1 contains python functions of only one method presented in the library, which might confuse the reader and reduce the quality of the flow of the paper. Because of all these considerations, we prefer scheme B1 to be part of Appendix B.

**References**

Amato, G., Palombi, L., and Raimondi, V.: Data–driven classification of landslide types at a national scale by using Artificial Neural Networks, International Journal of Applied Earth Observation and Geoinformation, 104, 102 549, 2021.

Loche, M., Alvioli, M., Marchesini, I., Bakka, H., and Lombardo, L.: Landslide susceptibility maps of Italy: Lesson learnt from dealing with multiple landslide types and the uneven spatial distribution of the national inventory, Earth-Science Reviews, p. 104125, 2022.

Marc, O., Hovius, N., Meunier, P., Gorum, T., and Uchida, T.: A seismologically consistent expression for the total area and volume of earthquake-triggered landsliding, Journal of Geophysical Research: Earth Surface, 121, 640–663, 2016.

Marc, O., Meunier, P., and Hovius, N.: Prediction of the area affected by earthquake-induced landsliding based on seismological parameters, Natural Hazards and Earth System Sciences, 17, 1159–1175, 2017.

Oksanen, J. and Sarjakoski, T.: Error propagation of DEM-based surface derivatives, Computers & Geosciences, 31, 1015–1027, 2005.

Ozturk, U., Pittore, M., Behling, R., Roessner, S., Andreani, L., and Korup, O.: How robust are landslide susceptibility estimates?, Landslides, https://doi.org/10.1007/s10346-020-01485-5, 2020.

Rana, K., Ozturk, U., and Malik, N.: Landslide Geometry Reveals its Trigger,
Geophysical Research Letters, 48, e2020GL090 848, 2021.

---

## Author Comment (AC2)

**Response to Reviewer 2**

September 13, 2022

We thank the reviewer for acknowledging our work, taking the time to read it, and giving suggestions to improve the work. We hope the reviewer will find our revised paper better suited for publication. We have highlighted changes to the paper in response to the reviewer's comments in blue in the revised article and in response to the reviewer's letter.

**Reviewer Comments on the Paper**

Dear authors,

I would like to express my appreciation on your work. I have read it with great interest and found the overall manuscript of particular scientific relevance for the geomorphology community.

Below I will summarize the manuscript content and later provide my feedback and suggestions, which I would like to stress here from the very beginning are very minor.

The manuscript deals with a very important topic, as it proposes a protocol to estimate the likely trigger of landslides from their shape and size characteristics. This is done by using a CNN architecture, which adds a numerical and methodological flavor to an article that adresses an important research question. In fact, as also stated by the authors, any attempt to predict landslides relies on previous information, this being usually expressed in polygonal inventories. However, often these inventories only provide the location of landlside occurrence and extent lacking to report the date. These inventories are usually geomorphological inventories (sensu Guzzetti et al. 2012) and they still represent the vast majority of the available inventories. This means that not knowing the date we cannot know the trigger responsible for the landslide occurrence, which is a fundamental requirement to then better understand the slope response over the whole affected landscapes.

Therefore, the protocol proposed by Rana and co-authors brings a very relevant tool for geomorphologists and for any other pratitioner, especially because of the way the authors opted to share their work through a python script. This is particularly important for repeatability and reproducibility.

Aside from this general overview of why I think this manuscript deserve to be published, specific elements support the same conclusion. In fact, the text

is extremely elegant and it flows nicely while reading it. I actually read it all in one go, which is something that not always occurs. In addition to the style and readability of it, I would like to stress the originality of the manuscript because to my knowledge at least, no other work has addressed the same issue, specifically with an open source solution to the problem. Also, the quality of scientific illustrations is very high. I am usually quite picky and yet I have no real comment to add, other that complimenting the authors.

We thank the reviewer for highlighting key innovations of our work. We hope landslide research community will find our work useful and further improve it.

In terms of feedback I can provide to improve the text, I have comments almost exclusively related to the literature review and what could be potentially added. One is that the authors refer to Taylor et al. (2018) in their text and rightfully so. But, at least for me a very similar if not better article has been recently published along the same lines that Taylor and co-authors introduced for the first time. The work I am mentioning is authored by Amato et al. (2021), where they also use a neural network architecture to explore landslide shape characteristics and infer on the landslide type at hand. This is even closer to your work because of the method they chose to use and I feel should be mentioned in your text. Another potential missing reference could be Lombardo et al. (2019), a paper where the trigger pattern has been derived from the inventory itself, although using a latent effect featured in a statistical model. Of course what you propose here is different but the question is basically the same: "How do I retrieve the trigger from the landslide themselves?". It is worth mentioning that I am the first author of that paper, therefore if you feel like I am imposing a reference, feel free to avoid my comment. I swear it is a genuine one, without a second interest to it.

Other than this minuscule details, your work is impecable to me and I would definitely be happy once I see it published in NHESS.

Again, congratulations. Kind regards,

Luigi Lombardo

References

Amato, G., Palombi, L. and Raimondi, V., 2021. Data–driven classification of landslide types at a national scale by using Artificial Neural Networks. International Journal of Applied Earth Observation and Geoinformation, 104, p.102549.

Guzzetti, F., Mondini, A.C., Cardinali, M., Fiorucci, F., Santangelo, M. and Chang, K.T., 2012. Landslide inventory maps: New tools for an old problem. Earth-Science Reviews, 112(1-2), pp.42-66.

Lombardo, L., Bakka, H., Tanyas, H., van Westen, C., Mai, P.M. and Huser, R., 2019. Geostatistical modeling to capture seismicâshaking patterns from earthquakeâinduced landslides. Journal of Geophysical Research: Earth Surface, 124(7), pp.1958-1980.

We appreciate the reviewer's comment about the work and feedback to improve it. We have included all the additional essential references in our main paper. Also, We have added a new paragraph in the Introduction section explaining more about the importance of trigger information for landslides. We reproduce the added paragraph below.

"Landslide planforms are used to estimate the mobilized landslide volume, for example, estimating the potential sediment budget of a large landslide triggering events (Malamud et al., 2004; Fan et al., 2012). This type of scaling relationship between the area of landslide planforms to mobilized landslide volume allows comparing the impact of different landslide triggers, such as human versus earthquakes, in terms of the landslides triggered influence on landscape (Tanyaş et al., 2022). However, this area-volume scaling depends on the triggering mechanism of landslides. For example, an earthquake-triggered landslide has a different area-volume relationship than a rainfall-induced landslide. Hence, extracting the landslide triggers information could enhance the estimation capacity of landslide volumes (Moreno et al., 2022) and also help predict the size of co-seismic landslides for a given earthquake (Lombardo et al., 2021). Also, when the exact trigger is known, observed landslides help assess earthquakes' ground motion patterns when no seismic observation is available (Lombardo et al., 2019)."

**References**

Fan, X., van Westen, C. J., Korup, O., Gorum, T., Xu, Q., Dai, F., Huang, R., and Wang, G.: Transient water and sediment storage of the decaying landslide dams induced by the 2008 Wenchuan earthquake, China, Geomorphology, 171, 58–68, 2012.

Lombardo, L., Bakka, H., Tanyas, H., van Westen, C., Mai, P. M., and Huser, R.: Geostatistical modeling to capture seismic-shaking patterns from earthquake-induced landslides, Journal of Geophysical Research: Earth Surface, 124, 1958–1980, 2019.

Lombardo, L., Tanyas, H., Huser, R., Guzzetti, F., and Castro-Camilo, D.: Landslide size matters: A new data-driven, spatial prototype, Engineering geology, 293, 106 288, 2021.

Malamud, B. D., Turcotte, D. L., Guzzetti, F., and Reichenbach, P.: Landslide inventories and their statistical properties, Earth Surface Processes and Landforms, 29, 687–711, 2004.

Moreno, M., Steger, S., Tanyas, H., and Lombardo, L.: Modeling the size of co-seismic landslides via data-driven models: the Kaikōura's example, https://doi.org/10.31223/x5vd1p, 2022.

Tanyaş, H., Görüm, T., Kirschbaum, D., and Lombardo, L.: Could road constructions be more hazardous than an earthquake in terms of mass movement?, Natural hazards, 112, 639–663, 2022.